# Ionic Liquid Transdermal Patches of Two Active Ingredients Based on Semi-Ionic Hydrogen Bonding for Rheumatoid Arthritis Treatment

**DOI:** 10.3390/pharmaceutics16040480

**Published:** 2024-04-01

**Authors:** Faxing Zhang, Lu Li, Xinyuan Zhang, Hongyu Yang, Yingzhen Fan, Jian Zhang, Ting Fang, Yaming Liu, Zhihao Nie, Dongkai Wang

**Affiliations:** Department of Pharmaceutics, School of Pharmacy, Shenyang Pharmaceutical University, No. 103, Wenhua Road, Shenyang 110016, China; zhangfx199901@163.com (F.Z.);

**Keywords:** rheumatoid arthritis, transdermal patch, semi-ionic H-bond, high drug loading, enhancing drug permeability

## Abstract

Rheumatoid arthritis (RA) is a chronic autoimmune disease that leads to deformities and disabilities in patients. Conventional treatment focuses on delaying progression; therefore, new treatments are necessary. The present study reported a novel ionic liquid transdermal platform for efficient RA treatment, and the underlying mechanism was elucidated using FTIR, ^1^H-NMR, Raman, XPS, and molecular simulations. The results showed that the reversibility of the semi-ionic hydrogen bonding facilitated high drug loading and enhanced drug permeability. Actarit’s drug loading had an approximately 11.34-times increase. The in vitro permeability of actarit and ketoprofen was improved by 5.46 and 2.39 times, respectively. And they had the same significant effect in vivo. Furthermore, through the integration of network pharmacology, Western blotting (WB), and radiology analyses, the significant osteoprotective effects of SIHDD-PSA (semi-ionic H-bond double-drug pressure-sensitive adhesive transdermal patch) were revealed through the modulation of the JAK-STAT pathway. The SIHDD-PSA significantly reduced paw swelling and inflammation in the rat model, and stimulatory properties evaluation confirmed the safety of SIHDD-PSA. In conclusion, these findings provide a novel approach for the effective treatment of RA, and the semi-ionic hydrogen bonding strategy contributes a new theoretical basis for developing TDDS.

## 1. Introduction

Rheumatoid arthritis (RA) is a chronic autoimmune disease that leads to deformities and disabilities in patients [1,2,3]. The current recommended treatment for RA includes nonsteroidal anti-inflammatory drugs (NSAIDs), glucocorticoids (GCs), disease-modifying anti-rheumatic drugs (DMARDs), and nanomedicine-based medication [4,5,6]. However, most single-pharmacological treatments focus only on relieving RA pain and do not address disease progression. Therefore, two-active-ingredient applications that relieve pain and inhibit RA progression are considered effective. DMARDs effectively inhibit the progressive damage to tissues and joints associated with RA, although they do not relieve pain. On the contrary, NSAIDs are often used to control pain and stiffness, although they do not slow disease progression. As a result, a combination of drugs is commonly used as an adjunctive treatment for RA and has been clinically validated [7,8,9,10,11]. Actarit is a DMARD that inhibits RA development of rheumatoid arthritis. However, its efficacy and bioavailability are limited owing to its low solubility, short half-life, and fluctuating blood concentration [12,13]. Ketoprofen is an NSAID commonly used to treat various types of pain [14]. To overcome the gastrointestinal side effects, the imperative need arises for a dosage form that complements this therapeutic approach.

Transdermal drug delivery systems (TDDSs) are widely recognized as effective for treating RA because they bypass the gastrointestinal side effects associated with oral administration and their potential to stabilize blood concentrations and enhance bioavailability [15]. Moreover, TDDSs offer the added advantage of convenience and targeted delivery to specific treatment sites, and transdermal patches are attracting attention owing to their simplicity of preparation and manufacturing ease [16,17,18,19], but the physiological barrier of the skin limits drug delivery [20,21,22,23]. Ionic liquids (ILs) can bypass the barrier properties of the outermost layer of the epidermis and diffuse intracellularly through mechanisms [24]. ILs are salts composed of cations and various anions formed by ionic interactions that typically remain in the liquid state below 100 °C [25]. Owing to their unique adjustable physical, chemical, and biological properties, cosolvents and materials are used as enhancers, counterions, and dosage-assisting forms to modify drug release or aid the synthesis of innovative materials for transdermal delivery [26,27]. It has been revealed that the fundamental mechanism of ILs is the presence of strong charge-assisted hydrogen bonds, van der Waals, and ionic interactions [28]. However, if these interactions become too strong, they can result in almost irreversible binding and reduced drug utilization [29]. In contrast, although weak and reversible, neutral hydrogen bonding minimizes drug loading [30]. Consequently, there is an urgent need to explore new interactions to overcome these drawbacks.

In order to investigate the potential therapeutic mechanisms of the two active ingredients for RA, we applied network pharmacology for screening, which was illustrated in combination with other experiments. Network pharmacology, which involves searching public databases and literature samples, is an effective method for understanding the mechanisms of drug action. This helps visualize potential drug and disease targets and predict drug targets and pathways [31]. 

Based on these factors, ionic liquid transdermal patches of two active ingredients, SIHDD-PSA, were developed to achieve effective analgesia and inhibit RA progression. Optimization of the formulation was carried out via Box–Behnken design. The action mechanisms were revealed by FTIR, ^1^H-NMR, Raman, XPS, and molecular docking. Network pharmacology and WB experiments demonstrated the potential role of actarit and ketoprofen in improving RA progression. Additionally, therapeutic efficacy and safety of SIHDD-PSA were demonstrated via in vivo/vitro animal studies and skin studies. The novel ionic liquid transdermal platform design offers a promising approach for achieving high drug loading and excellent permeability in TDDSs, as well as a potential pathway for the treatment of RA.

## 2. Materials and Methods

### 2.1. Materials and Animals

Actarit and ketoprofen were purchased from McLean Biochemicals (Shanghai, China); DURO-TAK^®^ 87-2287, 2852, and 4098 were purchased from Henkel AG (Dusseldorf, Germany); organic amine: triethylamine, methanolamine, diethanolamine, triethanolamine; permeation enhancers: azone, menthol, lauric acid, propanediol, N-methyl pyrrolidone, isopropyl myristate; and ethanol were purchased from Yuwang Pharmaceutical Co., Ltd. (Shenyang, China). All other chemicals were of reagent grade and purchased commercially.

Male Wistar rats (180–220 g) and Japanese White rabbits (1.5–2.5 kg) were obtained from the Laboratory Animal Center of Shenyang Pharmaceutical University. Animal experiments were performed in accordance with the Guidelines for Animal Experiments of Shenyang Pharmaceutical University and approved by the Animal Ethics Committee.

### 2.2. Synthesis of ILs

Actarit and ketoprofen were well-dispersed into acetone at a molar ratio of 1:1. After mixing thoroughly, equal molar amounts of triethylamine were added to the solution and stirred for 3 h at room temperature. The solvent was then removed by vacuum rotary evaporation at 30 °C. The resulting actarit–triethylamine–ketoprofen ionic liquid complex was dried in a vacuum oven at 30 °C for 24 h to remove the residual solvent.

### 2.3. Preparation of Transdermal Patches

The SIHDD-PSA were prepared via solvent evaporation [32]. The actarit–triethylamine–ketoprofen ionic liquid was dissolved in ethanol with an added permeation enhancer (azone, isopropyl myristate, etc.). The pressure-sensitive adhesive (PSA) was added to it until the clear solution appeared and allowed to stand for 10 min. The resulting solution was uniformly applied to the backing layer using an applicator, left at room temperature for 5 min, and then dried in an oven at 60 °C for 10 min. Finally, the sample was taken out and cooled to room temperature and covered with the backing layer (actarit 4.86 mg/cm^2^, ketoprofen 6.58 mg/cm^2^).

### 2.4. Drug Release 

The preparation of the isolated male Wistar rat skin was the same as described previously [33]. Release of drug from prepared patches was assessed using Franz diffusion setup. The acceptor compartment contained 4 mL of phosphate-buffered saline (PBS, pH 7.4). Samples were collected at specific time intervals (1 h, 2 h, 4 h, 6 h, 8 h, 10 h, 12 h, and 24 h) and were promptly replaced with fresh PBS to maintain consistent conditions throughout the experiment. The samples were analyzed using high-performance liquid chromatography (HPLC), and the relationship between the cumulative quantity (Q) of drug penetration per unit area and time (T) was plotted.

### 2.5. Optimization of Drug and Excipient Content

The formulation was screened via single-factor experiment and then the contents of formula component (including actarit, ketoprofen, triethylamine, and azone contents) were performed using Design Expert V8.0.6 software. A Box–Behnken four-factor three-level experiment was used to select the skin permeabilities of actarit and ketoprofen as the response values, and the important parameters of the two response values were set to the same value.

### 2.6. Fourier Transform Infrared Spectroscopy (FTIR)

FTIR spectroscopic studies can be used to assess the intermolecular interactions in patches [34]. The KBr disc coating method was used. Different drugs and their complexes, blank PSA and azone and PSA and azone samples loaded with the drug, were dissolved in ethanol and then added dropwise to the surface of the KBr discs, and the solvent was evaporated. Spectra were recorded using OPUS 7.2 software Bruker vertex 70 spectrometer, (Billerica, MA, USA). Each sample was scanned 16 times in the 4000–400 cm^−1^ wave number range. A blank KBr disk was used as the background.

### 2.7. Nuclear Magnetic Resonance Spectroscopy (^1^H NMR)

^1^H NMR spectroscopy was used to detect the chemical environment of the hydrogen atoms of the drug. The samples were dissolved in deuterated dimethyl sulfoxide as described above, placed in an NMR tube, and analyzed using an Advance 600 MHz spectrometer (Bruker, Benxi, China).

### 2.8. Raman Spectroscopy

Raman spectra were collected to study the molecular detail of ILs. Raman spectra were obtained using a Renishaw via laser Micro Raman spectrometer (Shanghai, China), equipped with an excitation laser operating at 785 nm with a laser power setting of 300 mW. The samples were placed on flat aluminum foil in front of a 50× objective lens. Raman spectra were acquired in the 4000–300 cm^−1^ spectral range with an acquisition time of 10 s for each spectrum. The spectral data were deconvoluted and analyzed for peak positions using Peak Fit 4.0 software. The final images were processed using WiRE software.

### 2.9. X-ray Photoelectron Spectroscopy (XPS)

XPS spectra was used to describe the amount of interaction. The samples were then dried and transferred to the XPS sample stage. XPS data were obtained using a Thermos ESCALAB XI+ instrument (Thermos Fisher Scientific, Waltham, MA, USA) with a monochromatic Al-Ka source (1486.6 eV, 150 W, 650 μm) for high-resolution spectroscopy and measurement spectroscopy at pass energies of 50 and 150 eV. The analytical process was performed with a pressure of 1 × 10^−10^ mbar, and the charge of each sample was corrected by referencing the C(1s) signal at 284.8 eV.

### 2.10. Molecular Docking

Molecular docking calculations confirmed the intermolecular interactions of the ILs. The structures of the drug molecules and triethylamine were searched using the PubChem database. Polymer chain structures were constructed based on the ratio of triethylamine to drug. The structures of the drug and triethylamine were geometrically optimized using the COMPASS II force field, and the resulting optimized structures were placed at appropriate positions to observe the potential sites and strengths of the semi-ionic hydrogen bonding interactions. 

### 2.11. Target Network Pharmacology Analysis

Actarit and ketoprofen standardized structures were obtained from the PubChem database (https://PubChem.ncbi.nlm.nih.gov/, accessed on 23 September 2023). These structures were imported into the Swiss Target Prediction database (http://www.wisstarget-prediction.ch/, accessed on 23 September 2023). The Swiss Target Prediction database helped predict potential targets for actarit and ketoprofen. To identify RA-related targets, the Gene Cards database (https://www.GeneCards.org/, accessed on 23 September 2023) was accessed using the keyword ‘rheumatoid arthritis’. The obtained drug and RA targets were organized and counted, and intersecting targets were screened. 

The shared targets were uploaded to the STRING online database (http://www.STRING-db.org, accessed on 23 September 2023) to establish a re-established drug target protein-disease target protein interaction network. The obtained results were imported into the Cystoscope 3.7.2 Software for mapping analysis. The DAVID database was used for gene ontology (GO) bifunctional annotation and the Kyoto Encyclopedia for Genes and Genomes (KEGG) for pathway enrichment analysis of intersecting core targets using the DAVID database. 

Crystal structures of the core targets were obtained from the PDB database for molecular docking validation. MAESTRO 4.6 software was used for the preprocessing, docking, and visualization of proteins and compounds.

### 2.12. Pharmacokinetics Study

The study randomly divided six male Wister rats into the actarit gavage and commercially available ketoprofen patch group and the SIHDD-PSA optimized patch group. Before applying the patches, hair on the abdomen of the rats was removed. The patches were then applied to the abdominal skin, with a surface area of 20 cm^2^ in the commercially available ketoprofen patch group and 3 cm^2^ in the optimized patch group (including approximately ketoprofen 20 mg). The actarit gavage dose was the same as the actarit content in the optimized patch (including approximately actarit 100 mg). At various time points (0.5, 1, 1.5, 2, 3, 4, 6, 8, 10, 12, and 24 h), 0.3 mL of blood was collected. Plasma samples were obtained by centrifugation at 16,000 rpm/min. For analysis, 100 μL of plasma samples were mixed with a 5 μL internal standard solution (methylparaben methanol solution, 100 μg/mL). The mixture was extracted using acetonitrile (1.0 mL) via vortex mixing for 3 min and centrifuged at 16,000 rpm for 5 min at 4 °C. The resulting organic layer was transferred to a test tube and evaporated at 40 °C using a stream of nitrogen. The residue was then redissolved in the mobile phase (50 µL) via vortex mixing for 2 min and centrifuged at 16,000 rpm for 5 min. Finally, a sample of 20 µL was injected into an HPLC system for analysis. Pharmacokinetic parameters were analyzed using the non-atrial model in WinNonlin^®^ 8.3.5 software [35].

### 2.13. Complete Freund Adjuvant–Induced Rheumatoid Arthritis

The anti-RA effect of the SIHDD-PSA was evaluated by measuring the swelling of rat feet. To induce joint damage, Complete Freund adjuvant (CFA, 0.1 mL) was injected into the tibiotarsal joint (−6 Day, The start period of our schedule is from −6 days continuing to 24 days, with measurements taken every three days). The procedure involved lightly anesthetizing the rats with ether, sterilizing the skin of the left leg with 75% ethanol, and inserting a syringe into the joint cavity at the gap between the tibiofibular and tarsal bone, where a noticeable decrease in resistance was observed [36]. The control group received an injection of 0.1 mL of saline solution in the tibiotarsal joint. The clinical signs of RA were characterized by swelling or redness of the paw and knee joints. The experimental groups were divided into the following: control, model, negative (AT), actarit–triethylamine patch (AAT), ketoprofen–triethylamine patch (KAT), oral actarit and commercially available ketoprofen patch (OAK), and homemade SIHDD-PSA patch (AKAT) groups. After the CFA administration, the knee of rats was treated daily starting from 0 day. The arthritis severity was evaluated using an Arthritis Scoring System (Arthritis Scoring System see Appendix A), and the circumference and thickness of both hind paws were measured every three days.

At the endpoint (Day 24), blood samples, synovial tissues, and intact knee joints of rats in each group were collected. The knee joints were fixed in paraformaldehyde and treated with a decalcification solution until decalcification was complete. Whole knees of the decalcification-treated rats were subjected to HE staining, and histopathological changes were observed after sectioning.

Bone erosion is an important factor in diagnosing and determining the severity of RA [37]. Foot-to-knee tissues were obtained at the endpoint for analysis via X-ray to determine the effect of patch treatment on bone destruction [38].

### 2.14. Western Blotting

The collected synovial tissue was added to the appropriate amount of RIPA lysis buffer, the total protein in the synovial membrane of the knee joint of the rats in each group was extracted, and the protein concentration was determined using the BCA method. After electrophoresing the 30 µg protein sample until the marker was layered, it was transferred to an activated polyvinylidene fluoride (PVDF) membrane. Following the transfer, the PVDF membrane containing the proteins and markers was sealed with 5% skimmed milk powder (prepared with TBST) and shaken on a shaker for 2 h. The membrane was then washed three times with TBST for 10 min each. Next, diluted primary antibodies including JAK2 (T55287, Purchased from Abmart, Shanghai, China, 10 µL), p-STAT3 (T56566, Purchased from Abmart, 10 µL), and STAT3 (T55292, Purchased from Abmart, 10 µL) were incubated at 4 °C overnight. The samples were removed and washed thrice with TBST for 10 min each. After incubation with the secondary antibody for 1 h, an ultrasensitive luminescent solution (liquid A:liquid B = 1:1) was prepared. The bands were detected and preserved using an FCM gel imager, with β-actin serving as a standard. The grayscale values of the protein bands were analyzed using ImageJ software.

### 2.15. Balance Beam Test and Imprinting Experiments

The balance beam test [39] involved placing rats on a narrow wooden board (30 cm × 1.3 cm) and scoring the ability of their bodies to maintain balance. Rats from the control, model, AT, AAT, KAT, OAK, and AKAT groups (three rats per group) were tested thrice. (For detailed scoring information, please refer to the Appendix A.)

After the rats were successfully modeled and treated for 12 days, the two hind feet were dipped into an appropriate amount of non-toxic carbon ink (black for the left foot and blue for the right foot) and placed into a box. The rats could pass through a narrow passageway to form a series of consecutive footprints. The walking footprints of the rats, which were mainly used as the indexes with which to detect the coordinated movements of the rats to evaluate the pain in terms of the width of the stride and the length of the stride, were recorded and statistically analyzed.

### 2.16. Skin Irritation and Barrier Function Study

An in vivo skin erythema study was performed to assess the irritant effects of SIHDD-PSA. The rabbits were divided into six groups: control, negative (AT group), positive (10% sodium dodecyl sulfate), AAT, KAT, K (commercially available ketoprofen patch), and AKAT. To ensure skin integrity, hair was removed from the abdomen 24 h before testing. First, the drug was administered to rabbit skin. The patch was removed, and the skin was washed with warm saline, and the degree of edema and erythema at the administration site was recorded at 12 and 24 h. Then, the abdominal skin of the rabbits was carefully excised, and the initial erythema value was measured as EI_0_ using a Mexameter (MX 16; Courage and Khazaka Co., Marseille, France), after which the patches were removed after 24 h and the EI was measured as a 24 h value.

To demonstrate the skin barrier function of the rat skin in each experimental group (experimental group as described above), the melanin and trans-epidermal water loss (TEWL) levels were measured in the isolated skin of rats. This was performed using a skin tester (MPA 9, Courage and Khazaka Co., Marseille, France).

### 2.17. Statistical Analysis

All results were expressed as mean values ± SD. Statistical analysis were performed by GraphPad 9.5.0 software (San Diego, CA, USA) using the analysis of variance test (ANOV A), followed by significance analysis with Sidaks’s multiple comparison test. Differences were considered significant at levels of * *p* < 0.05, ** *p* < 0.01 and *** *p* < 0.001, **** *p* < 0.0001. 

## 3. Results and Discussion

### 3.1. Effects of Counterions

The effect of counterions on the drug load of actarit was investigated as shown in Table 1, and this revealed an approximately 11.34-times increase in drug loads of actarit–triethylamine (visual checks of the samples were conducted to confirm the formation of ILs using the selected actarit and triethylamine; other counterions did not form ionic liquids). Figure 1a illustrates the ketoprofen findings. The skin permeabilities of the ILs exhibited the following trend: ketoprofen–triethylamine > ketoprofen–triethanolamine > ketoprofen–diethanolamine > ketoprofen–methanolamine. Ketoprofen–triethylamine displayed the highest skin permeability (159.23 ± 5.23 μg/cm^2^). Consequently, triethylamine was selected as the counterion for further investigations.

### 3.2. Effects of PSA

The drug loading capacity of adhesives containing different functional groups was investigated using three PSA. The results are shown in Table 2. The order of drug loading was DT-4098 ≥ DT-2852 > DT-2287. Considering the in vitro dermal release (Figure 1b,c), it was hypothesized that the presence of -OH in DT-2852 formed a strong ionic hydrogen bond with the drug and inhibited its release. This finding is consistent with the literature [40]. Based on these results, DT-4098 was selected as the PSA for this study.

### 3.3. Influence of Chemical Penetration Enhancers

According to Figure 1d,e, it is revealed that azone exhibited the highest penetration enhancement compared with the other enhancers. The penetration of actarit and ketoprofen was measured to be 234.66 ± 9.65 and 156.733 ± 5.64 μg/cm^2^, respectively. However, the effects of the remaining enhancers were not significant. Figure 1f showed that increasing the amount of azone did not significantly enhance the permeation of actarit or ketoprofen after reaching 6.25 mg/cm^2^. Therefore, azone was recommended for further screening of formulations. 

### 3.4. Content Optimization of Drugs and Excipients

The drug concentration influences drug permeability in the patch, as drugs passively diffuse through the skin. According to Fick’s law of diffusion, when the drug was not saturated, the permeability of drug increased as the drug concentration in the PSA increases. Hence, the present study aimed to achieve the maximum loading of actarit with an equimolar mass ratio of ketoprofen.

Based on the results of the screening, the patch composition was determined as follows: triethylamine as the counterion, DT-4098 as the PSA substrate, and azone as the permeation enhancer. The patch composition was further optimized using Design Expert V8.0.6 software. To maintain a high concentration gradient and stabilize the binding of the patch with drug loading, the loading of drug actarit (X_1_) and ketoprofen (X_2_) was set in the range of 1.00–5.00 mg/cm^2^ and 1.31–6.58 mg/cm^2^, respectively. The content of triethylamine (X_3_) was set in the range of 0.52–5.24 mg/cm^2^, the amount of azone (X_4_) ranged from 2.08–6.25 mg/cm^2^, and it was a commonly used range for permeation enhancers, which was considered the design space. The skin permeation amounts (Y_1_ and Y_2_) of actarit and ketoprofen were considered the response values, and the maximum value was set as the test target. A four-factor, three-level Box–Behnken design was used with the parameters listed in Table 3. The coded Box–Behnken design (actarit level (X_1_), ketoprofen level (X_2_), triethylamine level (X_3_), and azone level dosage (X_4_)) and the experimental results (responses Y_1_ and Y_2_) are listed in Table 4.

Finally, two coding factor equations were derived as follows:Y_1_ = 76.95 + 20.10*X_1_ − 6.55*X_2_ + 6.46*X_3_ + 19.99*X_4_ − 0.35*X_1_*X_2_ + 1.64*X_1_*X_3_ + 1.23*X_1_*X_4_ + 0.52*X_2_*X_3_ +, 0.34*X_2_*X_4_ + 1.12*X_3_*X_4_ − 2.76*X_1_^2^ + 0.58*X_2_^2^ − 2.56*X_3_^2^ − 2.67*X_4_^2^,
Y_2_ = −37.07 + 3.84*X_1_ + 22.42*X_2_ + 30.05*X_3_ + 5.19*X_4_ − 0.92*X_1_*X_2_ + 1.10*X_1_*X_3_ + 0.988*X_1_*X_4_ + 2.03*X_2_*X_3_ +, 1.52*X_2_*X_4_ + 1.85*X_3_*X_4_ − 1.23*X_1_^2^ − 1.72*X_2_^2^ − 6.54*X_3_^2^ − 1.38*X_4_^2^.

The significance test of the obtained model was presented in Figure 2, revealing *p* < 0.0001. The optimal formulation for the transdermal patch consisted of actarit 4.86 mg/cm^2^, ketoprofen 6.58 mg/cm^2^, triethylamine 2.54 mg/cm^2^ (with a molar ratio of approximately 1:1:1), and azone 6.25 mg/cm^2^. The final skin penetration resulted in 199.39 and 177.85 μg/cm^2^. These findings demonstrated that this method effectively enhanced drug loading and skin penetration.

### 3.5. Characterization of the Action Mechanism 

FTIR spectroscopy investigated the intermolecular interactions of two active ingredients ILs. Figure 3a showed that the -OH stretching vibrational peaks of actarit and ketoprofen were observed at 3331.44 and 3403.24 cm^−1^, respectively. The characteristic peaks disappeared upon deprotonation (actarit–triethylamine and ketoprofen–triethylamine). Notably, new peaks at 1538.55 and 1579.22 cm^−1^ were detected, which were attributed to the asymmetric stretching vibrations of COO^−^ resulting from the proton transfer between the carboxylic acid group of the drug and the amine group of the counterion [41]. These findings indicated the formation of ionic hydrogen interactions between actarit and triethylamine and between ketoprofen and triethylamine. Meanwhile, the ρ-OH peaks (in-plane bending vibrational peaks of -OH) of actarit–triethylamine and ketoprofen–triethylamine were observed at 1472.31 cm^−1^ and 1448.34 cm^−1^, respectively. In the actarit–triethylamine–ketoprofen system, the ρ-OH peak shifted to 1478.22 cm^−1^ and 1452.12 cm^−1^, suggesting that the in-plane rocking vibration of the -OH group was enhanced within the plane formed by the C, O, and H atoms. This indicated that the -OH group moved towards the distal end, weakening the ionic hydrogen interaction. The peak of -COO^−^ (R-O^−^) of ketoprofen–triethylamine shifted to a higher wave number due to the induced effect, which signified an increase in the R-O^−^ stretching frequency and further supported the weakening of the interaction. Furthermore, this ionic hydrogen bonding is not only present in actarit–triethylamine and ketoprofen–triethylamine. According to the study, chaining via H-bond between carboxylic groups occurs [42,43]. Thus, actarit–triethylamine–ketoprofen was a mixed complex between the three connected by semi-ionic hydrogen bonds. This special sub-stable interaction relationship leads to improved drug loading and drug permeation. Notably, the characteristic peaks did not show significant changes after loading with PSA or azone, indicating that the interactions were not affected by the presence of PSA or azone.

^1^H NMR spectroscopy was the most efficient method for characterizing these interactions. Figure 3b showed that the formation of actarit–triethylamine and ketoprofen–triethylamine resulted in a significant change in their ^1^H NMR spectra. The peaks representing the respective carboxyl groups (12.29 and 12.48 ppm) disappeared. In comparison, the chemical shifts of the hydroxypropyl groups near the carboxyl groups of the drugs decreased from 3.50 and 3.83 ppm to 3.43 and 3.69 ppm, respectively. This decrease was attributed to the weakening of the deshielding effect of COOH caused by the ionic interactions, resulting in an upfield shift. These findings indicated the formation of ionic hydrogen bonds. In the actarit–triethylamine–ketoprofen system, the shift of the methyne group to a lower field and the enhancement of the deshielding effect of COOH suggested the weakening ionic interactions, confirming the formation of semi-ionic hydrogen bonds consistent with the results obtained from FTIR spectroscopy.

Raman spectra was a valuable method for studying the molecular interactions in ionic liquids. In Figure 3c, the vibrational peaks corresponding to -OH coupling and C=O stretching in actarit and ketoprofen were observed at 1426.32, 1708.48 cm^−1^ and 1486.67, 1675.43 cm^−1^, respectively. However, in actarit–triethylamine and ketoprofen–triethylamine, the peaks at 1426.32 and 1486.67 cm^−1^ disappeared, and the C=O stretching peaks shifted to lower wave numbers (1693.46 and 1666.73 cm^−1^). This shift suggests that the -COOH groups in actarit and ketoprofen were completely deprotonated to -COO^−^. In contrast, in actarit–triethylamine-ketoprofen, the fully deprotonated -COO^−^ group shifted towards higher frequencies, indicating an enhanced vibrational frequency owing to a lower electron cloud density near the C-O^−^ region. These results confirmed the presence of semi-ionic hydrogen bonding interactions, consistent with the FTIR findings mentioned earlier.

XPS provided an electronic energy spectrum based on the photoelectric effect, which provided crucial information about these interactions. In Figure 3d, the O1s peaks for each group are analyzed. The O1s peaks at 532.8, 533.3, 539.6, and 533.5 eV were attributed to the -COOH groups of actarit–triethylamine, ketoprofen–triethylamine, and actarit–triethylamine–ketoprofen, respectively. Upon protonation, these -COOH groups were converted to -COO^−^, indicating the formation of ionic bonds between the drug and the counterion. In the actarit–triethylamine–ketoprofen system, after protonation, the -COO^−^ groups underwent ionization changes (43.86% to 28.37% and 46.55% to 26.49%) and the ionic interactions were reduced. The ability to maintain a relatively constant proportion of -COO^−^ in the system demonstrated its reversibility, and these results confirmed the formation of semi-ionic hydrogen bonds.

### 3.6. Molecular Docking

A molecular docking study was conducted to validate the interaction forced between the ionic liquids. As shown in Figure 4, the calculated distances between the -COOH and -NH groups of actarit–triethylamine and ketoprofen–triethylamine were 2.642 and 2.659 Å, respectively. In contrast, the distances of actarit–triethylamine–ketoprofen were 4.183 and 3.345 Å, indicating a weakening of the ionic interactions. Moreover, considering the energy aspect, actarit–triethylamine, ketoprofen–triethylamine, and actarit–triethylamine–ketoprofen binding energies were determined to be 37.65, 38.37, and 23.41 Kcal/mol, respectively. The observation strongly supported the formation of a stabilized semi-ionic hydrogen bond in actarit–triethylamine–ketoprofen.

### 3.7. Target Network Pharmacology Analysis

The 2D structure of the actarit and ketoprofen were generated using the ChemBioDraw Ultra 14.0 software. Actarit, ketoprofen, and RA targets were obtained from the Swiss Target Prediction and GeneCards databases to identify potential targets for treating RA. Figure 5a,b shows that Swiss Target Prediction predicted 101 and 100 targets for actarit and ketoprofen, respectively. The GeneCards database yielded 1423 RA-related targets. By comparing the targets of actarit and ketoprofen with RA-related targets, 21 overlapping targets were identified for actarit and 32 for ketoprofen. However, none of the overlapping ketoprofen targets were associated with a signaling pathway that inhibited bone injury. To investigate the inhibitory effect of actarit on bone erosion, we focused on studying its signaling pathway. The 21 overlapping targets were input into the STRING database to obtain the PPI network data. The resulting network, shown in Figure 5c, consists of 37 nodes representing interacting targets and 256 edges representing their interactions. Cytoscape 3.7.2 software was used for network analysis and visualization. The Network Analysis module provides attribute values, such as betweenness, closeness, and degree, for each node in the intersection network. By analyzing these values, we created a network diagram in which larger and darker nodes indicated higher importance, whereas darker lines represented closer relationships between the roles. The core targets identified were STAT3, ACE, CASP1, ITGAV, ALOX5, FABP4, MMP8, and so on. The values of the core target attribute are shown in Table 5.

Enrichment analysis was conducted using GO and KEGG to investigate the potential therapeutic mechanisms of the 21 putative actarit targets (Figure 5d,e). The size of the bubbles in the figure represents the number of genes enriched in each GO entry. In contrast, the color of the bubbles indicates the significance of the GO features, with redder bubbles representing smaller *p*-values. A total of 34 biological pathways (BPs), 14 cellular localizations (CCs), 10 molecular functions (MFs), and 7 KEGG pathways that met the criteria of p.adj < 0.1 and q value < 0.2 were identified. BP analysis revealed enrichment in biological processes such as inflammatory response, response to hypoxia, protein hydrolysis, positive regulation of cell proliferation, and microglia activation. Cellular localization analysis focused mainly on the cytoplasm, cytosol, and cytoplasmic membrane, whereas molecular function analysis highlighted protein binding, endopeptidase activity, and metallo-endopeptidase activity. The KEGG pathway enrichment analysis revealed closely related pathways to therapeutic targets, such as the JAK/STAT pathway, the cancer pathway, and the COVID-19 pathway of coronary viral disease. These pathways involved inflammation, metabolism, immunity, and other responses. By conducting actarit and ketoprofen target prediction and GO and KEGG pathway enrichment analyses, we hypothesized that actarit treatment of RA may be associated with regulatory pathways, including cell proliferation, cell survival, and regulation of inflammatory cytokines, particularly IL-17, rheumatoid arthritis, and TNF signaling pathways. Previous studies also reported a close relationship between these pathways, where synergistic interactions between IL-17 and TNF-α were shown to activate pro-inflammatory mediators such as IL-1β and IL-6. The JAK/STAT and RA signaling pathways played significant roles in the onset and progression of RA by mediating downstream inflammation. Therefore, based on these findings, the study focused on validating the JAK/STAT signaling pathway (Figure 6).

The node with the highest degree in the PPI network was selected as a pair acceptor. Crystal structures were obtained from the PDB database. Hydrogenation, dehydrogenation pretreatment, and visualization of the protein structure were performed using MAESTRO, and the results were shown in Figure 7. Actarit stably docked into the active pocket of the protein structure, indicating good binding between the two.

### 3.8. In Vivo Therapeutic Effect and Bone Assessment

The effect of SIHDD-PSA in rats with RA was evaluated by assessing arthritic swelling (the in vivo treatment process was shown in Figure 8d). Figure 8a showed that all treatment groups experienced varying degrees of relief from pathological symptoms. There was no significant change in swelling due to RA in the model and negative groups without drug treatment compared with the control group. Compared with the model group, the AAT group exhibited some relief from RA owing to the weaker anti-inflammatory effect of actarit. The KAT and AKAT groups significantly alleviated foot swelling and inflammation compared with the OAK group. The AKAT group demonstrated the most significant therapeutic effects. This indicated that the topical administration of SIHDD-PSA enhanced drug permeability, improved the drug release rate, thereby enhancing drug utilization, and consistently reduced inflammation. Severity scores and paw thickness assessments support these findings (Figure 8b,c), highlighting the synergistic treatment of the pathological progression of RA through the combination of DMARDs and NSAIDs.

The pharmacokinetic results are presented in Figure 8f. Compared with the commercially available ketoprofen patch, the SIHDD-PSA patch showed higher plasma concentrations of ketoprofen and an increased mean retention time. The oral actarit group exhibited a faster onset of action, with plasma concentrations rapidly decreasing after approximately 1 h, indicating rapid distribution and metabolism in the rat body. In contrast, the SIHDD-PSA patch group had a significantly longer mean retention time than the actarit gavage group. It peaked at approximately 3 h and plateaued from 3 to 6 h, suggesting successful temporal therapeutic drug delivery.

The model group exhibited significant synovial proliferation, inflammatory infiltration, and cartilage destruction. Histopathological analysis revealed that the application of ketoprofen in the KAT, OAK, and AKAT groups partially reduced inflammatory cell infiltration, and the application of actarit in the AAT, OAK, and AKAT groups inhibited synovial proliferation and cartilage erosion in Figure 9a. In the ionic liquid patch treatment group, minimal infiltration of inflammatory cells and minimal bone loss were clearly observed, suggesting that drug utilization can be improved by semi-ionic hydrogen bonding. The combined drug treatment of RA demonstrated a synergistic effect, and the AKAT group showed more pronounced relief of joint cavity lesions than the OAK group. The difference could be attributed to the high drug loading capacity and permeability of SIHDD-PSA.

X-ray analysis was conducted to confirm further and evaluate the inhibitory effect of SIHDD-PSA on bone and cartilage erosion. Figure 9a shows that the tarsometatarsal joint space was narrowed, the phalanges were destroyed, and there was swelling and high-intensity inflammation in the model and negative groups compared to the control group. However, the application of ketoprofen in the KAT, OAK, and AKAT groups significantly reduced swelling and inflammatory responses. Actarit treatment in the AAT, OAK, and AKAT groups also suppressed the narrowing of the tarsometatarsal joint space and erosion of the phalanges. Notably, the AKAT group demonstrated remarkable efficacy, with images comparable to those of the control group, as supported by pathology (Expressed at the red arrows in Figure 9a are the extent of bone erosion and joint gap size in the different groups.).

### 3.9. Western Blotting

Macrophage-associated cytokines in the inflamed joints played a crucial role in the pathogenesis of RA. Therefore, the study measured the serum concentrations of TNF-a and IL-6 in RA rats. As depicted in Figure 9b,c, KAT, OAK, and AKAT groups demonstrated inhibition of the up-regulation of TNF-a and IL-6. The AKAT group exhibited the strongest inhibitory effect, while the AAT group did not show a significant inhibitory effect, indicating that targeting these cytokines can effectively down-regulate macrophage function. Additionally, downstream proteins of the JAK-STAT pathway in the joints were analyzed using WB and normalized to β-actin. Figure 8e demonstrated a significant increase in phosphorylated STAT3 (*p*-STAT3) expression in RA rats, accompanied by pro-inflammatory TNF-a and IL-6 production. However, the OAK and AKAT groups exhibited attenuated STAT3 phosphorylation, with the AKAT group exhibited the most significant reduction. The AAT group also decreased STAT3 phosphorylation, although to a lesser extent compared to the changes in TNF-a and IL-6 levels. This could be attributed to the weaker anti-inflammatory effects of actarit. Similarly, the KAT group demonstrated a stronger anti-inflammatory effect but did not reduce STAT3 phosphorylation. These findings suggested that SIHDD-PSA effectively inhibited the JAK-STAT pathway, making it a potential therapeutic option for RA.

### 3.10. The Beam Balance Test and Imprinting Experiments

The balance beam test was used to evaluate locomotor injury. CFA injections impacted the animals’ motor coordination and walking ability, as shown in Figure 8h. Animals treated with drug-containing preparations showed reduced pathology scores, indicating partial recovery from locomotor injuries. AKAT exhibited the most significant therapeutic effect.

The footprints of the rats are shown in Figure 9d. Compared with the control group, the model group exhibited pronounced incoordination caused by pain and foot swelling. The resulted in shorter and wider strides in the mice. However, pain and coordination issues were partially alleviated in the drug-treated group, leading to increased step length and narrower stride width. Interestingly, rats in the AKAT group demonstrated significantly larger and narrower strides (Figure 9e,f), indicating a substantial alleviation of pain and the best recovery achieved in the AKAT group.

### 3.11. The Skin Irritation and Barrier Function Experiment

Erythema tests were performed to assess skin irritation caused by SIHDD-PSA. As shown in Figure 8g, the skin of the positive group showed significant erythema and edema and a significant increase in EI value to 4.20 ± 0.10. In contrast, the other groups were normal, and the EI value did not change significantly (*p* > 0.05); both results proved that SIHDD-PSA was safe.

The results indicated that the positive control group exhibited decreased melanin content and an increased transdermal water loss, suggesting a decline in the skin barrier function (Figure 9g). However, no significant changes were observed in the barrier function of the skin in the other experimental groups.

## 4. Conclusions

In this study, a two-active-ingredients transdermal patch was developed, and the mechanism of semi-ionic hydrogen bond was discovered. The findings suggested that the formation of the interaction influenced the rejection of drug–ILs–drug, resulting in an approximately 11.34-times increase in actarit drug loading. The permeability of actarit and ketoprofen was increased by 5.46 and 2.39 times, respectively. The mean retention times and plasma concentrations of actarit and ketoprofen were significantly higher in the SIHDD-PSA group than in the control group. Furthermore, network pharmacology analysis has shown that actarit downregulates the JAK-STAT pathway, thereby slowing the progression of RA. In contrast, ketoprofen did not directly target the progression of RA but reduced inflammatory factors, thus alleviating pain. Through this study, we can find that actarit in combination with ketoprofen (or other NSAIDs) was more effective in inhibiting RA than each drug alone, and this application was widely used in the clinic and holds the promise of achieving the desired therapeutic effect in clinical applications. In vivo studies demonstrated that SIHDD-PSA effectively reduced foot swelling and inflammation in rats, showing similarities to the normal group. Notably, SIHDD-PSA not only alleviates pain, as anticipated, but also mitigates bone damage. Skin irritation tests confirmed the gentle application of SIHDD-PSA on the skin. Thus, SIHDD-PSA represented a reasonable, effective, and safe transdermal patch. The discovered semi-ionic hydrogen bonding interactions offer promising drug delivery and permeability capabilities, suggesting a potential strategy for the long-term clinical treatment of RA and other diseases.

## Figures and Tables

**Figure 1 pharmaceutics-16-00480-f001:**
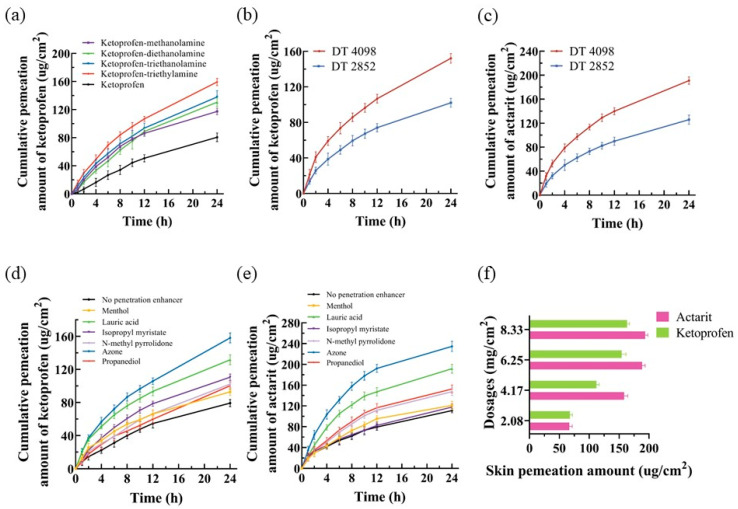
In vitro skin permeation for (**a**) ketoprofen and its ionic liquid complexes patches; (**b**,**c**) different PSA patches of ketoprofen and actarit ionic liquid complexes; (**d**,**e**) ketoprofen and actarit ionic liquid complexes patches with different penetration enhancer; and (**f**) ketoprofen and actarit with different levels of azone as permeation enhancers (*n* = 3).

**Figure 2 pharmaceutics-16-00480-f002:**
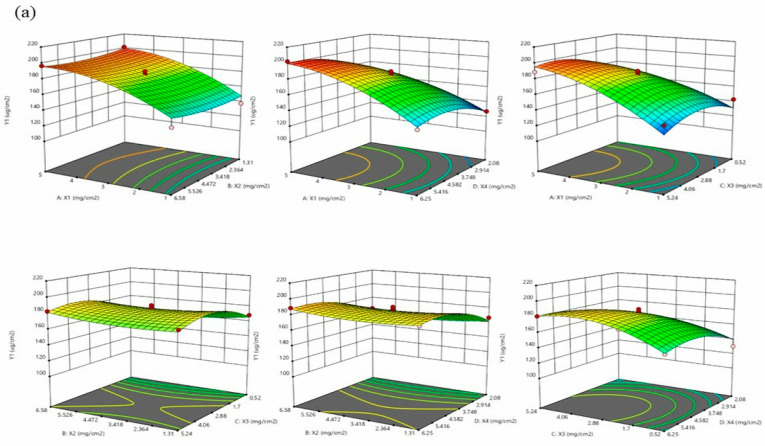
Prescription optimization response surface. (**a**,**b**) Actarit and ketoprofen content X_1_ and X_2_, triethylamine content X_3_, and azone content X_4_ (mg/cm^2^) to the response Y_1_ and Y_2_ (μg/cm^2^) of skin permeation amount.

**Figure 3 pharmaceutics-16-00480-f003:**
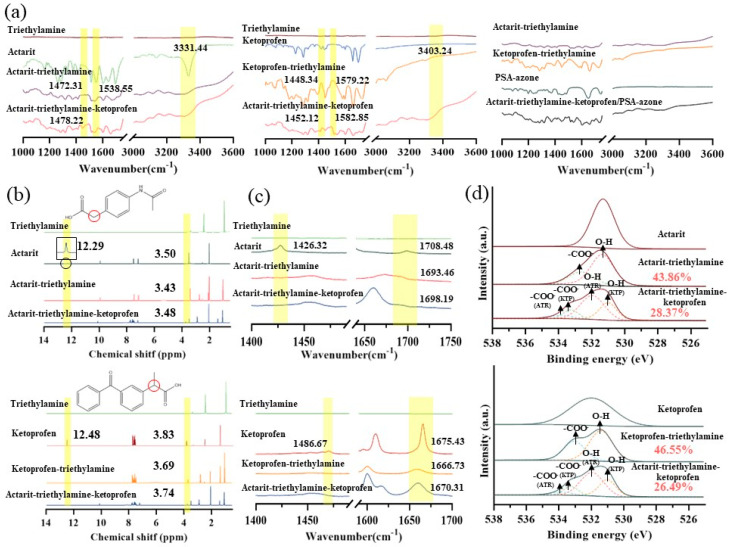
Characterization of drug(actarit)–counterion(triethylamine)–drug(ketoprofen) interactions in ILs. (**a**) FTIR spectra of actarit and ketoprofen and their ionic liquid complexes. (**b**) Chemical structures of actarit and ketoprofen, and ^1^H NMR spectroscopic analysis of the chemical shifts of actarit, ketoprofen, and their ionic liquid complexes. (**c**) Raman spectra. (**d**) COO^−^ and O-H peaks obtained from XPS.

**Figure 4 pharmaceutics-16-00480-f004:**
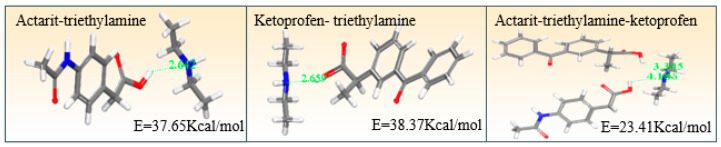
Molecular docking of actarit–triethylamine, ketoprofen–triethylamine, and actarit–triethylamine–ketoprofen.

**Figure 5 pharmaceutics-16-00480-f005:**
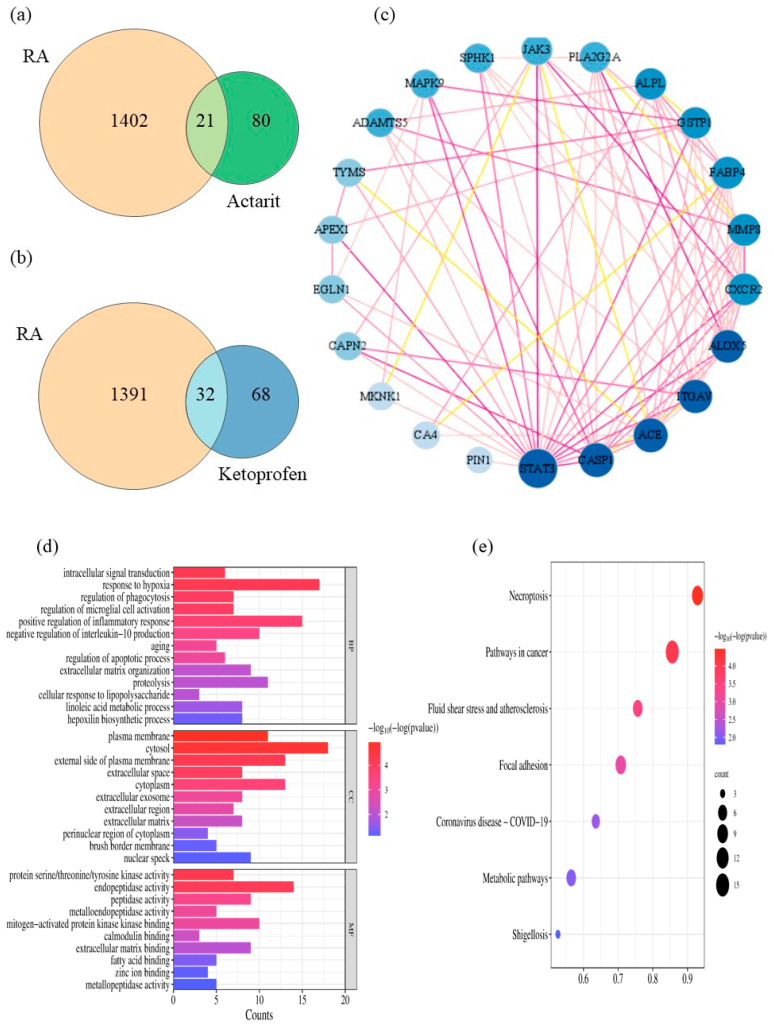
Acquisition and network analysis of common targets of the drugs in treating RA. (**a**,**b**) Venn diagrams of potential targets of actarit, ketoprofen, and RA. (**c**) Target protein interaction network diagram. (**d**,**e**) GO enrichment and KEGG pathway enrichment analyses.

**Figure 6 pharmaceutics-16-00480-f006:**
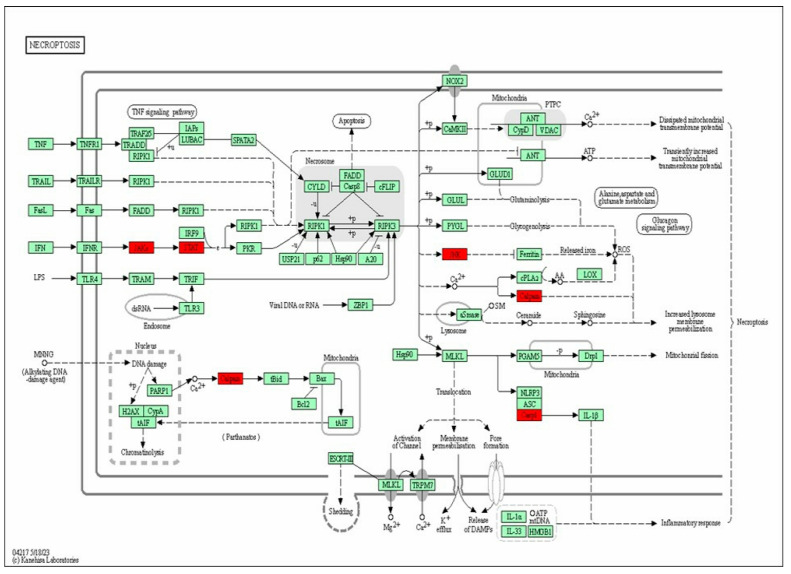
JAK/STAT signaling pathway mapper. The arrows (⟶) represent the promoting effect, the tarrows represents the inhibiting effect, and the enriched genes were marked with red star symbols.

**Figure 7 pharmaceutics-16-00480-f007:**
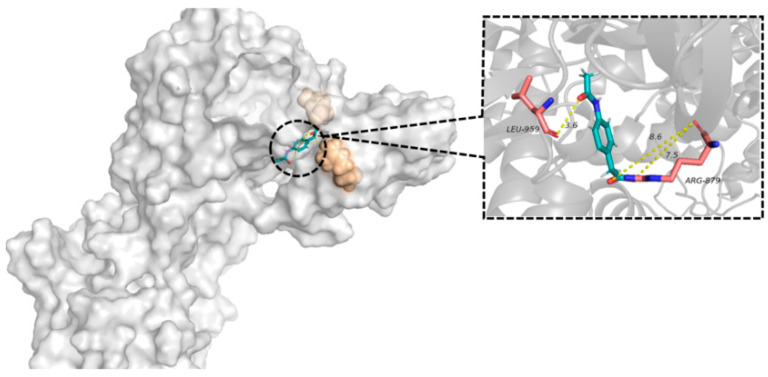
Actarit and JAK-STAT molecular docking simulations.

**Figure 8 pharmaceutics-16-00480-f008:**
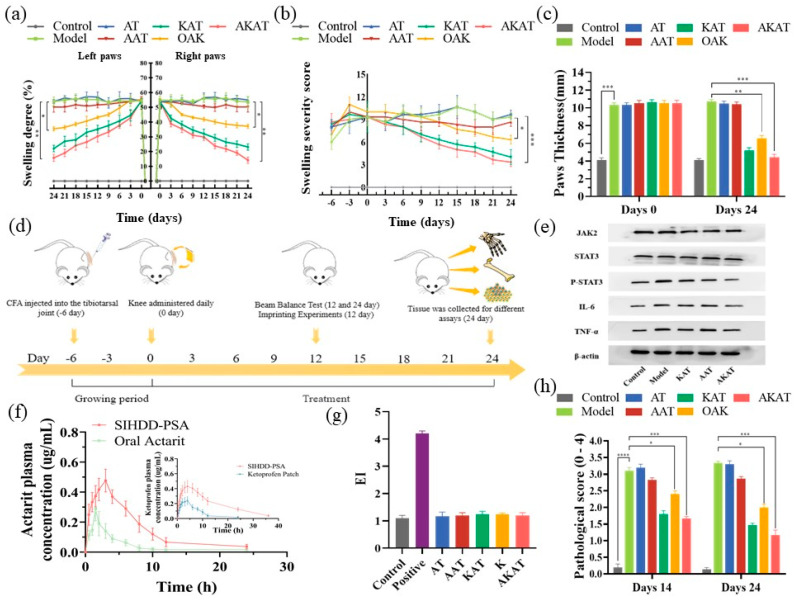
In vivo therapeutic effects of SIHDD–PSA: (**a**) the profiles of swelling degree (%) of paws after CFA injection; (**b**) assessment of severity scores during treatment in rats; (**c**) changes in paw thickness in rats after treatment; (**d**) animal experimentation schedule; (**e**) Western blotting analysis for expression of JAK2, *p*−STAT3 and STAT3 in arthritic joints; (**f**) ctarit and ketoprofen plasma concentration–time curves; (**g**) EI values for each group; (**h**) motor abilities related to pain. (*n* = 3 and * *p* < 0.05, ** *p* < 0.01 and *** *p* < 0.001, **** *p* < 0.0001). The uncropped blots are shown in Appendix A.

**Figure 9 pharmaceutics-16-00480-f009:**
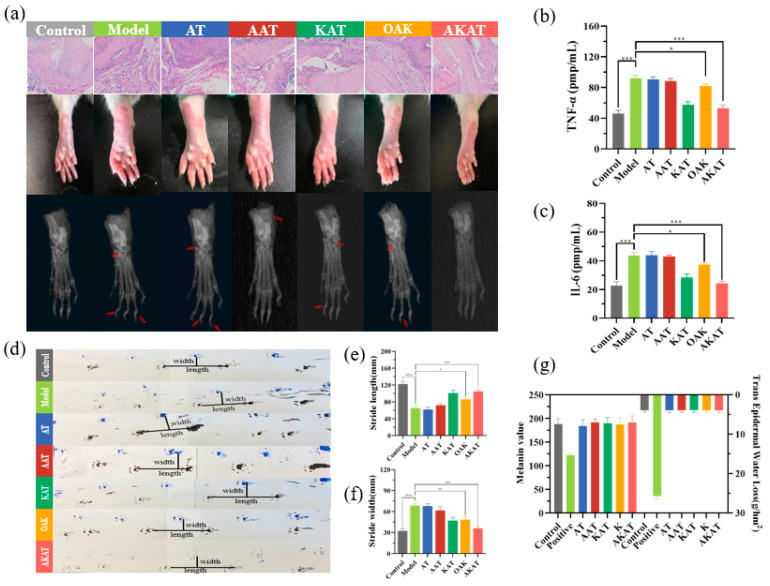
Anti-inflammatory treatment of SIHDD-PSA: (**a**) HE staining and hind paw X-ray images of knee joint sections from various groups; (**b**,**c**) concentrations of TNF-a and IL-6 in serum of rats from each group in steps; (**d**) footprints on the hind limbs of rats in each group; (**e**,**f**) statistical analysis of stride length and stride width in rats; (**g**) melanin and transcutaneous water loss content express skin barrier functions. (*n* = 3 and * *p* < 0.05, ** *p* < 0.01 and *** *p* < 0.001, **** *p* < 0.0001).

**Table 1 pharmaceutics-16-00480-t001:** Screening results for maximum drug loading of different ILs in actarit.

Counterions	Methanolamine	Diethanolamine	Triethanolamine	Triethylamine	No
Drug loading (mg/cm^2^)	1.00	1.00	1.00	5.00	0.44

**Table 2 pharmaceutics-16-00480-t002:** Screening results of maximum drug loading of three kinds of pressure-sensitive adhesives.

PSA	DT 4098	DT 2852	DT 2287
Drug loading of actarit (mg/cm^2^)	5.00	5.00	1.00
Drug loading of ketoprofen (mg/cm^2^)	6.58	6.58	2.63

**Table 3 pharmaceutics-16-00480-t003:** Factor codes and levels of Box–Behnken design.

Level	Actarit Content (X_1_, mg/cm^2^)	Ketoprofen Content (X_2_, mg/cm^2^)	Triethylamine Content (X_3_, mg/cm^2^)	Azone Content (X_4_, mg/cm^2^)
+1	1.00	1.31	0.52	2.08
0	3	3.94	2.88	4.16
−1	5.00	6.58	5.24	6.25

**Table 4 pharmaceutics-16-00480-t004:** Encoding Box–Behnken design (drug, counterion content, and permeation enhancer content X_1_, X_2_, X_3_, X_4_, respectively) and experimental results (response Y_1_ and Y_2_ of drug skin permeation).

Run	X_1_	X_2_	X_3_	X_4_	Y_1_	Y_2_
1	0	0	0	0	181.64	124.94
2	0	−1	+1	0	177.48	58.97
3	0	+1	−1	0	160.18	87.14
4	0	0	0	0	182.65	152.01
5	−1	+1	0	0	140.31	179.42
6	−1	0	−1	0	142.79	73.43
7	+1	0	+1	0	188.74	128.94
8	0	0	0	−1	130.19	52.15
9	−1	0	0	+1	138.17	140.98
10	0	+1	−1	−1	164.45	156.47
11	0	0	−1	+1	152.68	67.54
12	0	−1	+1	0	168.18	44.68
13	−1	0	+1	0	142.64	115.77
14	0	0	0	-1	136.21	103.15
15	0	0	0	0	175.18	138.23
16	+1	0	0	−1	170.89	97.15
17	0	0	0	0	177.65	138.54
18	0	−1	0	−1	166.64	59.87
19	0	+1	0	+1	188.62	195.64
20	+1	−1	0	0	201.25	70.53
21	0	0	0	0	185.64	127.13
22	0	−1	0	+1	183.12	65.63
23	−1	0	0	−1	127.19	105.48
24	+1	+1	0	0	196.56	167.89
25	0	+1	+1	0	182.65	152.01
26	−1	−1	0	0	137.59	62.48
27	0	0	+1	+1	180.94	155.14
28	+1	0	−1	0	157.79	65.74
29	+1	0	0	+1	202.49	149.14

**Table 5 pharmaceutics-16-00480-t005:** Core target properties.

Target	Betweenness Centrality	Closeness Centrality	Degree
STAT3	150.09	0.95	20
ACE	37.33	0.72	13
CASP1	22.21	0.72	13
ITGAV	15.51	0.70	12
ALOX5	7.91	0.68	11
FABP4	11.54	0.66	10
MMP8	5.19	0.07	10

## Data Availability

The data presented in this study are contained within the article.

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
