# Peer review of "Ionic Liquid Transdermal Patches of Two Active Ingredients Based on Semi-Ionic Hydrogen Bonding for Rheumatoid Arthritis Treatment"

_pharmaceutics, 2024, doi:10.3390/pharmaceutics16040480_

Round 1

Reviewer 1 Report

Comments and Suggestions for Authors

  The article submitted by Zhang et al deals with the preparation of ionic liquid transdermal patches of Actarit and Ketoprofen based on semi-ionic hydrogen bonding for rheumatoid arthritis treatment. The topic is relevant in the field and explores alternative delivery systems for effective therapy in rheumatoid arthritis. The manuscript is well structured, but there are some points to be addressed by the authors, which are:

1.     Methodology, drug release; How did the authors maintain sink condition? The solubility of drugs tested is likely to be low in the PBS used.

2.     Which skin membrane was used in the drug release study?

3.     The HPLC method mentioned in the supplementary section is incomplete. Please include the retention time and validation data.

4.     Are both drugs analyzed simultaneously? Please provide more information.

5.     Why do the authors do Target Network Pharmacology Analysis? The pharmacological activity of these 2 drugs is well-known. Explain.

6.     Figure 1, the permeation profiles of both drugs look rapid with a very short lag time. Explain.

7.     Did the authors check the permeation of these drugs separately? A comparison of data could have been interesting. The skin permeability of the permeant is limited and the co-delivery is likely to decrease the inherent permeation of the solute.

8.     Did the authors observe any drug crystallization in the prepared patches?

9.     There is room for improvement in the discussion section. The authors typically described the results and the observations are not discussed in any sections.

10.  Add a statement related to the clinical relevance of this observation.

Author Response

Dear editors and reviewers:

Thank you very much for your careful review and constructive suggestions about our manuscript ID pharmaceutics-2908671, " Ionic liquid transdermal patches of two active ingredients based on semi-ionic hydrogen bonding for rheumatoid arthritis treatment". Those comments are helpful for authors to revise and improve our paper. We have studied the comments carefully and tried our best to revise according to the reviews′ comments. The revised portion is marked in red in the paper. The main corrections in the paper and the response to the reviewers ′comments are as following:

Thanks again for your kind directions. The careful, detailed comments are extremely helpful for us to revise our manuscript. Best regards.

Yours sincerely,

Dongkai Wang

Comments and Suggestions for Authors

  The article submitted by Zhang et al deals with the preparation of ionic liquid transdermal patches of Actarit and Ketoprofen based on semi-ionic hydrogen bonding for rheumatoid arthritis treatment. The topic is relevant in the field and explores alternative delivery systems for effective therapy in rheumatoid arthritis. The manuscript is well structured, but there are some points to be addressed by the authors, which are:

  1. Methodology, drug release; How did the authors maintain sink condition? The solubility of drugs tested is likely to be low in the PBS used.

Thanks to the reviewer for this careful comment. Some studies have shown that the solubility of the drug can be improved after the preparation of the drug into ionic liquid. Our previous pre-experiment found that the solubility of the two drugs after the formation of ionic liquid was indeed improved, meeting the sink condition.

  1. Which skin membrane was used in the drug release study?

Thank you very much for your valuable comments, we used the abdominal skin of male Wistar rats in the drug release study and it was revised in the manuscript.

  1. The HPLC method mentioned in the supplementary section is incomplete. Please include the retention time and validation data.

Thank you for pointing out our problems, the retention time of actarit and ketoprofen is 5.23 min and 6.33 min, validation data is shown in word, and we have added the missing data to the supplementary document. 

Figure 1 The HPLC chromatograms of actarit(left) and ketoprofen(right).

  1. Are both drugs analyzed simultaneously? Please provide more information.

Thank you for your suggestions and comments. Our drugs are not analyzed simultaneously, it is measured by two methods. Under the same conditions of detection wavelength, sample size and others, the mobile phase of methanol: acetic acid solution (0.5%, v/v) ratio of actarit is 30:70(v/v). The mobile phase of methanol: acetic acid solution (0.5%, v/v) ratio of ketoprofen is 70:30(v/v). The chromatogram is the same as 3.

  1. Why do the authors do Target Network Pharmacology Analysis? The pharmacological activity of these 2 drugs is well-known. Explain.

Thanks to the reviewer for this careful comment. During our search, we found that the description of actarit in the original research manual and other materials is "possible" to achieve therapeutic effects through some effects, the exact mechanism of the drug is unclear and this drug is not currently listed in China. Ketoprofen is a non-steroidal anti-inflammatory drug whose pharmacological activity is well known. Ketoprofen was also screened to rule out the effect of ketoprofen on actarit, but it was found that ketoprofen did not affect the experiment, so this study focused on actarit. We did this experiment for the accuracy of the experiment.

  1. Figure 1, the permeation profiles of both drugs look rapid with a very short lag time. Explain.

Thanks to the reviewer for this careful comment. First, this preparation has a higher loading capacity and therefore provides a higher concentration difference during transmission. The translucent agent we use (azone) penetrates into the stratum corneum of the skin, disrupts the arrangement of cellular lipids, reduces the action of the skin barrier, and accelerates the passage of drugs through the skin. The ionic liquid complex we formed improved the physical and chemical properties of the drug, had a more suitable Log P value, was better able to penetrate the skin, and the ionic liquid also played a certain role in promoting penetration.

  1. Did the authors check the permeation of these drugs separately? A comparison of data could have been interesting. The skin permeability of the permeant is limited and the co-delivery is likely to decrease the inherent permeation of the solute.

We would like to thank the reviewer for these constructive comments. We checked the penetration of these drugs separately, and there was no significant difference between co-delivery (permeability of actarit and ketoprofen was 234.67±9.65 and 158.07±6.07 μg/cm2) and separate delivery of drugs (227.70±9.27 and 151.99±7.98 μg/cm2) from the penetration data. The opinions of reviewers also provide a very important reference value for our research. In subsequent studies, we may investigate the inherent permeability of the solute.

  1. Did the authors observe any drug crystallization in the prepared patches?

Thank you for the helpful comments. No drug crystallization were observed in the ionic liquid transdermal patch under a microscope. Below is a comparison of non-ionic liquid and ionic liquid transdermal patches (the little white dots on the right are not drug crystals but bubbles that enter during preparation).

Figure 2 Non-ionic liquid transdermal patches (left), ionic liquid transdermal patches(right)

  1. There is room for improvement in the discussion section. The authors typically described the results and the observations are not discussed in any sections.

Thanks to the reviewer for this careful comment. We have improved our discussion of experimental results and phenomena and would like to express our thanks again.

  1. Add a statement related to the clinical relevance of this observation.

Thank you very much for your valuable comments. We have added the clinically relevant narrative and would like to express our thanks again.

Finally, I would like to sincerely thank the editors and reviewers again for their valuable suggestions, and thank you for spending your precious time to give us guidance.

Reviewer 2 Report

Comments and Suggestions for Authors

Development of new transdermal transport medications is presented in manuscript. It is important area with great research and application interest. Wide valuable experimental material with deeper insight to mechanism of drug activity and structuring is presented. Presentation of results is in some parts messy and needs revision before manuscript could be recommended for publication. Authors may find useful considering some comments and suggestions in upgrading their manuscript or in future work.

1)      There are used numerous abbreviations in manuscript, some of them are not explained. This makes its reading difficult. For example abbreviation SIHDD-PSA is used 25 times in manuscript but not explained. In L. 85 azetidine (AZ) is mentioned among permeation enhancers, but in Fig. 1 is mentioned Azone what is different compound. What is permeation enhancers AZ mentioned in sections 3.3 and 3.4? What is “permeation enhancer (AZ IPM et al.)” presented in L. 103?

2)      Section 2.1: It is messy and fragmental presentation of used chemicals and materials used. Chemicals should be presented chemical names not by abbreviations or trade names, especially for key compounds like ATR, KTP and permeation enhancer also full structural formula. Group of ionic liquid (IL) precursors and permeation enhancers should be listed separately to show their role. Group of DURO-TAK´s are pressure-sensitive adhesives (PSA).

3)       Amonium and phosphonium ILs extract carboxylic acids and IL/acid molar ratio in related complexes can achieve values much larger than 2 as shown for extraction of butyric acid with phosphonium IL [1]. Chaining via H-bond between carboxylic groups occurs. This suggests, that when to equimolar solution of ATR with TEtA is added KTP they can form mixed complexes TEtA-ATR-KTP where H-bonds between carboxylic groups of ATR and KTP play role. In light of this scheme in top left side insert of the graphical abstract could be misleading. Uncomplete structural formulas of ATR and KTP are presented. Complexity of system follows also from discussion in paper [2].

4)      Reference 32: Author names are incorrectly cited. It should be Song, H. instead of Haoyuan S., etc. for other co-authors.

 [1] Marták, J., Liptaj, T., and Schlosser, Š., Extraction of Butyric Acid by Phosphonium Decanoate Ionic Liquid, Journal of Chemical & Engineering Data, 64 (2019) 2973−2984. 

[2] Berton, P., Kelley, S.P., Wang, H., and Rogers, R.D., Elucidating the triethylammonium acetate system: Is it molecular or is it ionic?, Journal of Molecular Liquids, 269 (2018) 126. 

Author Response

Dear editors and reviewers:

Thank you very much for your careful review and constructive suggestions about our manuscript ID pharmaceutics-2908671, " Ionic liquid transdermal patches of two active ingredients based on semi-ionic hydrogen bonding for rheumatoid arthritis treatment". Those comments are helpful for authors to revise and improve our paper. We have studied the comments carefully and tried our best to revise according to the reviews′ comments. The revised portion is marked in red in the paper. The main corrections in the paper and the response to the reviewers ′comments are as following:

Thanks again for your kind directions. The careful, detailed comments are extremely helpful for us to revise our manuscript. Best regards.

Yours sincerely,

Dongkai Wang

Comments and Suggestions for Authors

Development of new transdermal transport medications is presented in manuscript. It is important area with great research and application interest. Wide valuable experimental material with deeper insight to mechanism of drug activity and structuring is presented. Presentation of results is in some parts messy and needs revision before manuscript could be recommended for publication. Authors may find useful considering some comments and suggestions in upgrading their manuscript or in future work.

1) There are used numerous abbreviations in manuscript, some of them are not explained. This makes its reading difficult. For example abbreviation SIHDD-PSA is used 25 times in manuscript but not explained. In L. 85 azetidine (AZ) is mentioned among permeation enhancers, but in Fig. 1 is mentioned Azone what is different compound. What is permeation enhancers AZ mentioned in sections 3.3 and 3.4? What is “permeation enhancer (AZ IPM et al.)” presented in L. 103?

We are very sorry for these problems. We have explained all abbreviations while reducing the use of abbreviations and correcting the material and thank the reviewer for taking the valuable time to raise these issues to us. We have carefully checked the problems in the manuscript from the beginning to the end and corrected them.

2) Section 2.1: It is messy and fragmental presentation of used chemicals and materials used. Chemicals should be presented chemical names not by abbreviations or trade names, especially for key compounds like ATR, KTP and permeation enhancer also full structural formula. Group of ionic liquid (IL) precursors and permeation enhancers should be listed separately to show their role. Group of DURO-TAK´s are pressure-sensitive adhesives (PSA).

Thank you very much for your valuable comments, we have summarized and classified the ionic liquid precursors, penetration promoters and pressure-sensitive adhesives in Section 2.1. Don't use abbreviations to describe important chemicals and would like to express our thanks again.

3) Amonium and phosphonium ILs extract carboxylic acids and IL/acid molar ratio in related complexes can achieve values much larger than 2 as shown for extraction of butyric acid with phosphonium IL [1]. Chaining via H-bond between carboxylic groups occurs. This suggests, that when to equimolar solution of ATR with TEtA is added KTP they can form mixed complexes TEtA-ATR-KTP where H-bonds between carboxylic groups of ATR and KTP play role. In light of this scheme in top left side insert of the graphical abstract could be misleading. Uncomplete structural formulas of ATR and KTP are presented. Complexity of system follows also from discussion in paper [2].

We would like to thank the reviewer for this important comment. Thank you very much for your idea, which provides a very meaningful guidance for our research. Due to the formation of hydrogen bonds between carboxyl groups, it is indeed possible for these three compounds to form a co-existing complex based on semi-ionic hydrogen bonds, which we did not think of, we feel particularly ashamed and revise the summary diagram in the manuscript.

4) Reference 32: Author names are incorrectly cited. It should be Song, H. instead of Haoyuan S., etc. for other co-authors.

We would like to thank the reviewer for this important comment. We have revised all the references and we are sorry for our oversight.

[1] Marták, J., Liptaj, T., and Schlosser, Š., Extraction of Butyric Acid by Phosphonium Decanoate Ionic Liquid, Journal of Chemical & Engineering Data, 64 (2019) 2973−2984.

[2] Berton, P., Kelley, S.P., Wang, H., and Rogers, R.D., Elucidating the triethylammonium acetate system: Is it molecular or is it ionic?, Journal of Molecular Liquids, 269 (2018) 126.

Finally, I would like to sincerely thank the editors and reviewers again for their valuable suggestions and thank you for spending your precious time to guide us.

Round 2

Reviewer 1 Report

Comments and Suggestions for Authors

None

Author Response

Dear editors and reviewers:

Thank you very much for your careful review and constructive suggestions about our manuscript ID pharmaceutics-2908671, " Ionic liquid transdermal patches of two active ingredients based on semi-ionic hydrogen bonding for rheumatoid arthritis treatment". 

With your help, our research has qualitatively improved. Thank you for your support and all the authors express their most respectful gratitude to you!

Yours sincerely,

Dongkai Wang

Reviewer 2 Report

Comments and Suggestions for Authors

Authors reflected comments and suggestions of reviewer in revised manuscript. Upgraded manuscript can be suggested for acceptation. Authors possibly forgot to insert the revised summary diagram as mentioned in reply to comment 3.

Suggestion for future work:

A mixed three component complex in present work was prepared in two steps, first TEtA-ATR and later KTP was added to get TEtA-ATR-KTP. From reference 42 follows that the IL-solute H-bond is much more stable than the second solute-solute H-bond. Thus, more distant solute from IL will be more easily deliberated from the mixed complex. In actual system TEtA-ATR bond comparing to ATR-KTP bond. Probably, positions of ATR and KTP in chain will change. But, it could be of interest study if order in which components of mixed complex are added when prepared or when both ATR and KTP will be added at once to TEtA will influence the final efficiency of the transdermal patch.

Author Response

Dear editors and reviewers:

Thank you very much for your careful review and constructive suggestions about our manuscript ID pharmaceutics-2908671, " Ionic liquid transdermal patches of two active ingredients based on semi-ionic hydrogen bonding for rheumatoid arthritis treatment". Those comments are helpful for authors to revise and improve our paper. We have studied the comments carefully and tried our best to revise according to the reviews′ comments. The revised portion is marked in red in the paper. The main corrections in the paper and the response to the reviewers ′comments are as following:

Thanks again for your kind directions. The careful, detailed comments are extremely helpful for us to revise our manuscript. Best regards.

Yours sincerely,

Dongkai Wang

Authors reflected comments and suggestions of reviewer in revised manuscript. Upgraded manuscript can be suggested for acceptation. Authors possibly forgot to insert the revised summary diagram as mentioned in reply to comment 3.

Suggestion for future work:

A mixed three component complex in present work was prepared in two steps, first TEtA-ATR and later KTP was added to get TEtA-ATR-KTP. From reference 42 follows that the IL-solute H-bond is much more stable than the second solute-solute H-bond. Thus, more distant solute from IL will be more easily deliberated from the mixed complex. In actual system TEtA-ATR bond comparing to ATR-KTP bond. Probably, positions of ATR and KTP in chain will change. But, it could be of interest study if order in which components of mixed complex are added when prepared or when both ATR and KTP will be added at once to TEtA will influence the final efficiency of the transdermal patch.

Author's reply:

We are very ashamed that our negligence led to this problem. The idea you proposed is especially important for this paper, which largely improved this paper and gave us new ideas, the H-bond between ATR and KTP is more stable than ATR-TEtA and KTP-TEtA, and it is in accordance with the conditions of the literature 42, thank you again for the reviewer's suggestion. We re-uploaded the images and modified the preparation method.

Graphical Abstract

Finally, I would like to sincerely thank the editors and reviewers again for their valuable suggestions and thank you for spending your precious time to guide us.
